# Synthetic promoters to induce immune-effectors into the tumor microenvironment

Yariv Greenshpan[1,2], Omri Sharabi[1,2], Aner Ottolenghi[1,2], Avishag Cahana[1,2], Kiran Kundu[1,2], Ksenia M. Yegodayev[1], Moshe Elkabets[1], Roi Gazit [1,2,3✉] & Angel Porgador[1,2,3✉]

Harnessing the immune-system to eradicate cancer is becoming a reality in recent years. Engineered immune cells, such as chimeric antigen receptor (CAR) T cells, are facing the danger of an overt life-threatening immune response due to the ON-target OFF-tumor cytotoxicity and Cytokine Release Syndrome. We therefore developed synthetic promoters for regulation of gene expression under the control of inflammation and Hypoxia-induced signals that are associated with the tumor microenvironment (TME). We termed this methodology as chimeric-antigen-receptor-tumor-induced-vector (CARTIV). For proof of concept, we studied synthetic promoters based on promoter-responsive elements (PREs) of IFNγ, TNFα and hypoxia; triple PRE-based CARTIV promoter manifested a synergistic activity in cell-lines and potent activation in human primary T-cells. CARTIV platform can improve safety of CAR T-cells or other engineered immune-cells, providing TME-focused activity and opening a therapeutic window for many tumor-associated antigens that are also expressed by non-tumor healthy tissues.

[1] Faculty of Health Sciences, The Shraga Segal Department of Microbiology, Immunology and Genetics, Ben-Gurion University of the Negev, Beer Sheva, Israel. [2] National Institute for Biotechnology in the Negev, Ben-Gurion University of the Negev, Beer Sheva, Israel. [3] These authors contributed equally: Roi Gazit, Angel Porgador. ✉email: gazit@bgu.ac.il; angel@bgu.ac.il

Cancer treatment is a major challenge facing modern medicine. Cancer cells originate from the patient own healthy cells, thus any treatment targeting the cancer cells might also harm healthy tissues. Thus, there is an ongoing pursuit for finding new treatments that will target exclusively cancer cells manifesting minimal to zero toxicity to healthy tissues. Precision oncology, developed to increase efficacy and reduce toxicity, usually involves the molecular profiling of tumors identifying targets that discriminate tumor from healthy tissues; while immunotherapy aims to manipulate the immune system in to specifically attacking the cancer cells[1]. One technique that combines both approaches is chimeric antigen receptors (CARs). CAR T cells treatments were granted already approvals by the FDA, with first success treatments targeting CD19 in refractory B-cell malignancies. To date, there are over 400 ongoing clinical trials regarding CAR therapy. Nonetheless, CAR therapy success has also highlighted its potential treatment-related toxicities and it is thus currently limited due to the residual expression of the studied targets on healthy tissues mediating the on-target-off-tumor toxicity[2–6].

Some innovative strategies have been studied to overcome or reduce this on-target-off-tumor toxicity. One approach taken is introducing a suicide gene in to the T cell. In this approach, when the inducible caspase 9 (iCasp9) can be dimerized and leads to rapid apoptosis of T cells expressing the iCasp9 suicide gene by addition of the synthetic dimerizing drug, AP1903[7,8]. A different approach is based on inhibition: an antigen recognition domain with specificity to antigens only presented on normal tissue is fused to an inhibitory receptor. Thus, the CAR T cells can distinguish the tumor and non-tumor cells and restrict their activity to the tumor domain[9]. Another strategy for reduced toxicity is based on splitting the activation signal of the CAR T cell. In this method the full activation signal is divided to two CARs, each recognizing a different antigen, and each is carrying only part of the TCR signaling/stimulation domain e.g. one carrying the CD3ζ and the other is carrying the CD28/4-1BB[10,11]. Hence, full activation is mediated by the presence of both antigens that are tumor specific are recognized the full activation potential of the T cell will be realized. Another intriguing approach to reduce ON target OFF tumor activity has been proposed by Juillerat et al. where they fused an oxygen sensitive subdomain of HIF1α to a CAR scaffold to generate self-decision making CAR T cell sensitive to the TME hypoxic niche[12]. This approach is of great interest but it relays on a single factor to induce the CAR.

We approached the on-target-off-tumor toxicity problem by restricting spatiotemporal CAR expression through coupling the CAR gene to a synthetic inducible promoter that is sensitive to combined stimuli portraying the tumor microenvironment (TME). TME is the environment around a tumor, including the surrounding blood vessels, immune cells, fibroblasts, signaling molecules and[13–17], the extracellular matrix[18–22]; the TME is often hypoxic[23–26], demonstrating a characteristic pro-tumorigenic inflammation mediated by both tumor and non-tumor cells of the TME. Hence, we designed synthetic promoters that are (i) combined from promoter-responsive elements (PREs) responding to various TME-associated cytokine stimuli and hypoxic conditions, and (ii) constructed to optimally respond to the combination of stimuli activating the promoter PRE's building blocks. We define this technology as CARTIV: chimeric antigen receptor tumor-induced vector. Specifically, we show that a 3-PRE promoter composed from PREs responding to IFNγ, TNFα, and hypoxic stimuli can be employed to restrict CAR expression to the tumor site.

## Results

**CARTIV design of promoter and reporter construct**. Based on consensus Promoter-Response-Elements (PRE) sequences for IFNγ, TNFα or hypoxia[27–29], we first designed three CARTIV-adapted PREs (CPREs): GCPRE, KCPRE, and HCPRE for IFNγ, TNFα or hypoxia, respectively (Fig. 1a). To generate CARTIV promoters, the different CPREs, separated by linkers, were then homo- and hetero-combined in a "mix and match" manner, upstream to a minimal herpes virus thymidine kinase[30] (mini TK). Complete sequences of these CPREs and the tested CARTIV promoters are summarized in Supplementary Fig. 1. Figure 1b shows a scheme of 2-CPRE- and 3-CPRE-combined CARTIV synthetic promoters.

To test the activity of the CARTIV synthetic promoters, we cloned them upstream to a fluorescent protein reporter RFP670 in a 3rd generation lentiviral vector. Our vector also harbored another independent constitutive promoter driving a distinct ZsGreen fluorescent reporter (Fig. 1c), allowing to easily identify the infected cells using flow cytometry and to assess the CARTIV promoter activity.

**2-CPRE CARTIV promoters are responsive to IFNγ and TNFα**. To functionally test CARTIV promoter's activities in cells, we needed a model that will be responsive to IFNγ, TNFα, and hypoxia. HEK293T cells express IFNγ and TNFα receptors and are responsive to human IFNγ and TNFα in a wide range of concentrations (Supplementary Fig. 2A–C), and are responsive to hypoxia[31,32]. To validate the elements functionality, we examined the single GCPRE and KCPRE elements as individual promoters (Supplementary Fig. 3) as well as CARTIV synthetic promoter combined from GCPRE and KCPRE; we tested the following promoters: one KCPRE (K1); two, four, and six GCPRE (G2, G4, G6, respectively); three GCPRE and three KCPRE (G3K3); two GCPRE upstream to two KCPRE (G2K2); one GCPRE upstream to one KCPRE (G1K1), and one GCPRE upstream to a short version of the KCPRE sequence (G1K0.6). Background and maximum expression were defined by expression of the RFP670 reporter from a mini TK promoter alone with no added elements and a full length EF1α promoter, respectively (Supplementary Fig. 2D). CARTIV+ HEK293 cells were incubated with either 250 U/mL IFNγ, 250 U/mL TNFα or both for 48 h and analyzed by FACS for reporter expression. CARTIV promoter activity was

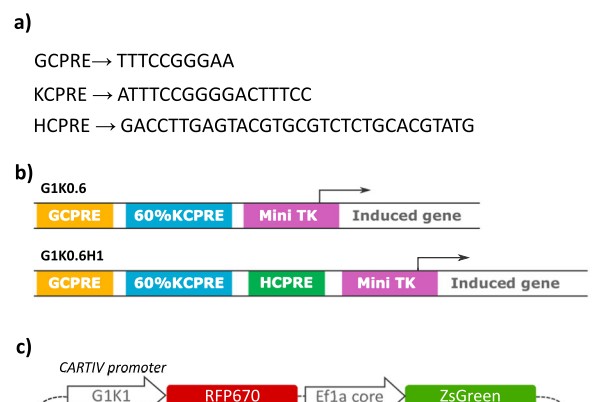

**a)**

GCPRE→ TTTCCGGGAA

KCPRE → ATTTCCGGGGACTTTCC

HCPRE → GACCTTGAGTACGTGCGTCTCTGCACGTATG

**b)**

G1K0.6

| GCPRE | 60%KCPRE | Mini TK | Induced gene |

G1K0.6H1

| GCPRE | 60%KCPRE | HCPRE | Mini TK | Induced gene |

**c)**

CARTIV promoter

G1K1 → RFP670 – Ef1a core → ZsGreen

**Fig. 1 The promotor response elements used to construct the CARTIV promotes and the vector used. a** CARTIV promotor response elements (CPREs) used in CARTIV promotors. **b** Two of the tested promotors. Yellow: IFNγ CPRE (GCPRE), turquoise: NFkB CPRE (KCPRE), green: hypoxia CPRE (HCPRE), purple: minimal herpes simplex virus thymidine kinase (Mini TK). **c** The vector used in the CARTIV system. Direct activation of the CARTIV vector, including an independent constitutive reporter.

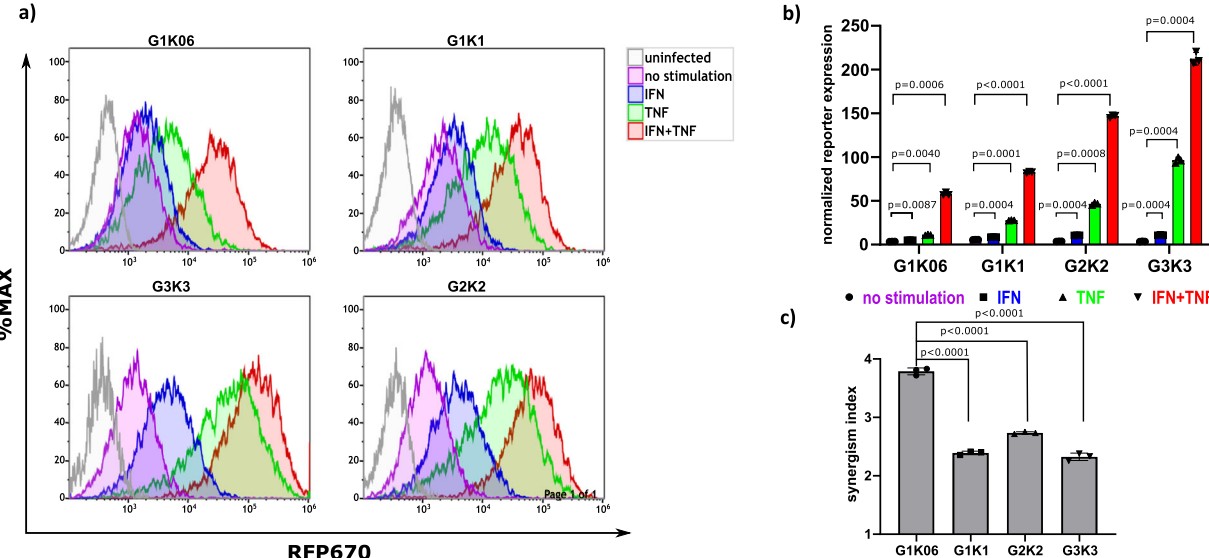

**Fig. 2 CARTIV promotors show a robust and additive effect with IFNγ and TNFα stimulation in HEK293 cells. a** Representative FACS plots of HEK293T cells infected using lenti viral vectors with RFP670 under the control of the indicated CARTIV promotor and ZsGreen controlled by the ef1α core promoter. At 72 h following infection, the cells were incubated for 48 h with the indicated cytokines (250 U/mL for each cytokine), harvested and analyzed by flow cytometry. Data shown are ZsGreen-positive, single-discriminated, and DAPI-negative results. **b** Geometric mean of RFP670 in ZsGreen-positive cells normalized to uninfected cells, showing average of triplicates, error bars indicate standard deviation, A two way ANOVA was performed. **c** Synergism level calculated by dividing the geometric mean of ZS+ cells stimulated with IFN and TNF by the geometric mean of cells stimulated with IFN or TNF. A one-way ANOVA was performed, error bars indicate standard deviation. Results are from one representative experiment of three full experiments performed (including the G2K2); the G1K0.6, G1K1, and G3K3 were compared in more than eight independent experiments.

assessed by gating on ZsGreen+ cells (the transduced cells) and testing the geoMEAN of the RFP670 reporter (Supplementary Fig. 4). The K1 promoter manifested a strong activity as a single element (73.38 fold increases compared to non-stimulated) when compared to the GCPRE that showed no substantial difference in expression (2.87, 6.38, and 6.43 fold increases for G2, G4, and G6 accordingly when compared to non-stimulated) between 4 and 6 GCPRE promoters (Supplementary Fig. 3). Following stimulation with IFNγ alone, the GCPRE manifested low activity for G1K0.6 and G1K1 and higher activity for the G2K2 and the G3K3 (1.35, 1.57, 3.16, and 3.47 fold increases for G1K0.6, G1K1, G2K2, and G3K3 accordingly when compared to non-stimulated) as seen in Fig. 2a. Following stimulation with TNFα alone, the KCPRE manifested a pronounced activity that maximized in the G3K3 promoter (2.9, 5.01, 13.88, and 32.82 fold increases for G1K0.6, G1K1, G2K2, and G3K3 accordingly when compared to non-stimulated) as seen in Fig. 2a. As expected, a single-factor reactivity was increasing with the number of repeats; nevertheless, the most synergistic activation following a combined stimulation was best observed for the G1K0.6 promoter induction by IFNγ and TNFα (Fig. 2c). The CARTIV promoters having more repeats showed pronounced activation (16.13, 15.75, 46.61, and 84.33 fold increases for G1K0.6, G1K1, G2K2, and G3K3 accordingly when compared to non-stimulated), but their synergistic response was relatively lower, likely due to the strong induction by TNFα alone (Fig. 2c).

**The hypoxia element better location is immediately upstream to the mini TK.** Gaining good activation by one inflammatory factors and even better with synergistic two factors, we next set out to find a third signal to combine in our CARTIV promoters. In solid tumors it is common to find hypoxia due to insufficient vascularization and blood supply[23–26]. Therefore, we designed CARTIV promoters combined from GCPRE, KCPRE and also HCPRE for hypoxia-response (Fig. 3a). We first aimed to find if there might be a preferential location of the HCPRE within the

CARTIV promoter. A series of three promoters were constructed to consist of G, K, and H elements that differ only in the location of the hypoxia element relative to the minimal TK (Fig. 3a). CARTIV+ HEK293T subjected to hypoxic conditions for 18 h and ZsGreen+ cells were assessed for the expression of the RFP670 reporter. Interestingly, all three promoters responded to the hypoxia stimuli, a 1.69, 2.07, and 3.91 fold increase compared to the normoxic conditions for the H2G2K2, G2H2K2, and the G2K2H2 accordingly; however, the response was most profound when the HCPRE was located downstream to the other CPREs and adjacent to the minimal TK (Fig. 3b, c). This finding was unanticipated since the HRE consensus sequence is usually found between 100 and 1000 bases upstream to the ATG start codon[29] and may appear also downstream of the TSS[33]. Hence, our data demonstrate critical importance for the order of PRE elements within a CARTIV promoter that is not predictable and require experimental examination.

**G1K06H1 promoter is responsive to triple-stimuli in lower cytokine concentrations.** Following the above mentioned findings for double GCPRE-KCPRE promoter (Fig. 2) and the location of the HPRE (Fig. 3), we designed the G1K06H1 synthetic promoter with one copy of GCPRE, 60% of the KCPRE, and one HCPRE. CARTIV+ HEK293 cells were incubated with 500 U/ml IFNγ and 500 U/ml TNFα for 24 h then placed under hypoxic or normoxic conditions for an additional 18 h; in this experimental setup the hypoxia stimuli did not add to the activation induced by IFNγ and TNFα; Initial experiments with 500 U/ml of each cytokine found little effect by the addition of hypoxia (Fig. 4a). We further investigated this promoter in lower levels of cytokine stimuli that better correlates with the physiological conditions in the TME[17,34–36]. Cells were incubated for 24 h with all possible combinations of 500, 125, 32, 8, 2, and 0 U/ml of IFNγ and TNFα, and then placed under hypoxic or normoxic conditions for an additional 18 h (Fig. 4b, c). Data clearly indicates that in the lower concentration of IFNγ and TNFα, the added effect of the hypoxia

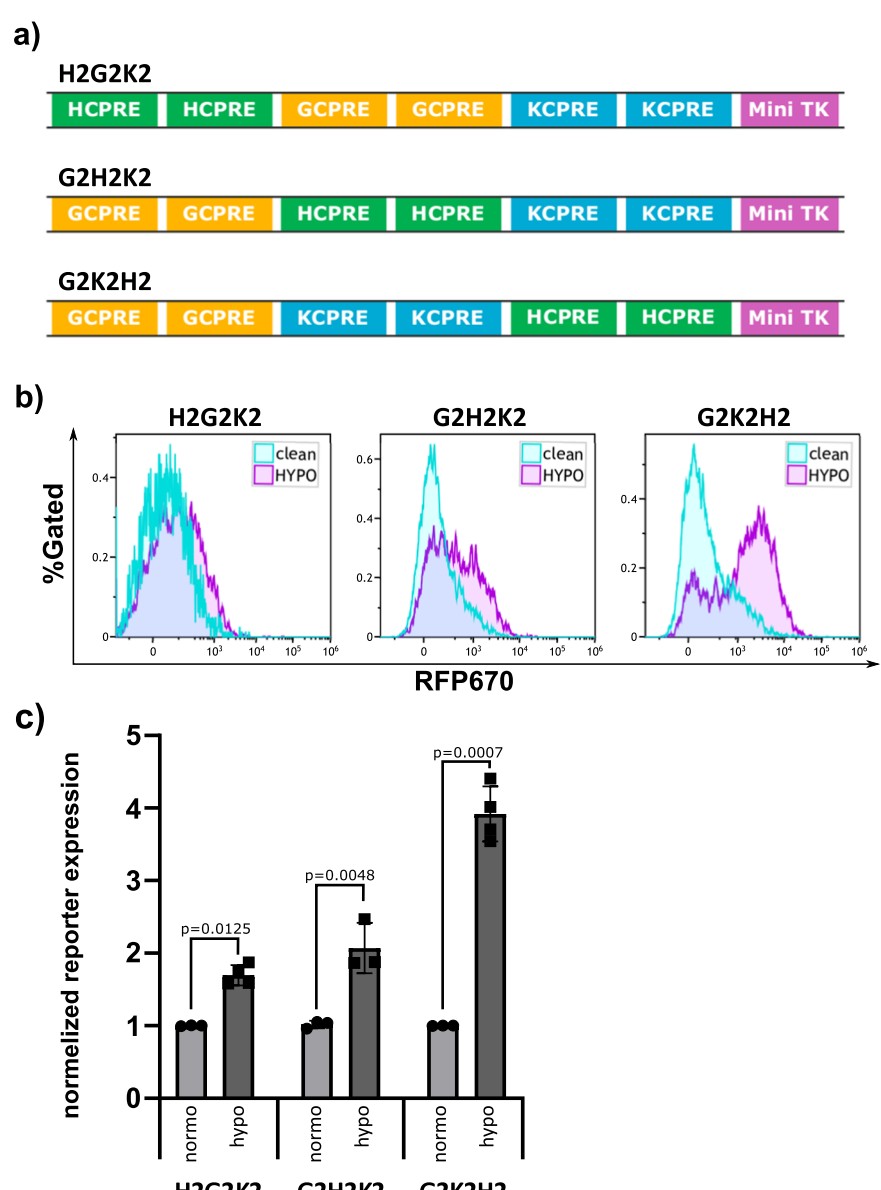

**Fig. 3 The hypoxia PRE in CARTIV promotors is responsive and most potent when upstream of the minimal promotor. a** Schematic drawing showing the different hypoxia element locations relative to the mini TK minimal promotor tested. **b** Representative plots of HEK293T cells infected using lenti viral vectors with RFP670 under the control of the indicated CARTIV promotor and ZsGreen controlled by the ef1α core promotor. At 72 h following infection, the cells were placed under hypoxic conditions, incubated for 18 h, harvested and analyzed by flow cytometry. Data shown are ZsGreen-positive, single-discriminated, and DAPI-negative results. **c** Geometric mean of RFP670 in ZsGreen-positive cells normalized to unstimulated cells, showing the average of duplicates, error bars indicate standard deviation. A *t* test was performed to test for significance. Results are from one representative experiment of four performed.

stimulus on the RFP670 reporter expression is more profound. Figure 4d shows the synergistic induction by fold-expression following the hypoxia stimulus; when both IFNγ and TNFα stimuli were lower than 32 U/ml the fold induction due to addition of the hypoxia stimulus was the highest. The "turn on" and "turn off" kinetics of the G1K06H1 promoter were tested by adding 500 U/mL of IFNγ and TNFα and tracking the increase or decrease in florescence over 48 h. The promoter showed a $t_{1/2}$ of 9.78 h on rate and a $t_{1/2}$ of 12.37 h off rate (Supplementary Fig. 5).

**G1K06H1 promoter in NK92 cells and primary human T cells is responsive to combined TNFα and hypoxia stimuli.** Following the experiments with the HEK293T cell line, we aimed to investigate the activity of our CARITV promoter in immune effector cells that are clinically relevant to CAR-based treatment. We used primary human T cells as well as the NK92 natural killer cell line[37]. Following stable transduction of the G1K0.6H1 CARTIV, cells, primary human T cells or the NK-92 cell line, were stimulated with 250 U/ml of IFNγ, TNFα, and Hypoxia. Both NK92 and primary human T cells manifested a substantial response to the combined stimuli of hypoxia and TNFα as compared to controls (a 5.206 and 3.84 fold increase compared to the non-stimulated for the NK92 and human primary T cells accordingly), and to single-factor stimulation (Fig. 5a, b). Interestingly, in this set of experiments IFNγ stimulus did not show substantial nor significant effect with either T- or NK-cells (Fig. 5b). This relative low impact of IFNγ was probably due to the levels of endogenous IFNγ secreted by these cultured T or NK cells. Direct measure of IFNγ from these cells cultures indeed

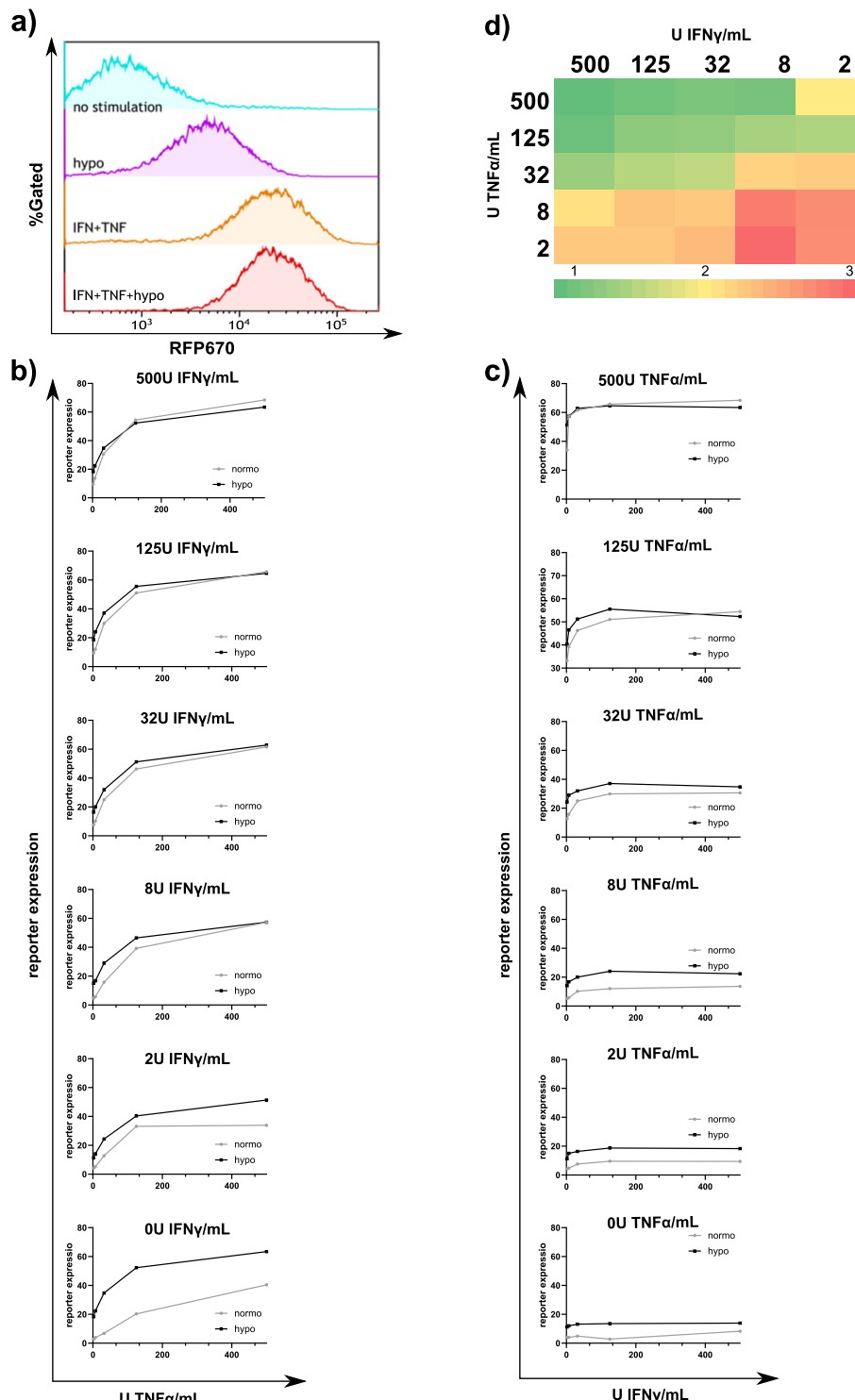

**Fig. 4 The G1K06H1 promotor shows an additive response to hypoxia and cytokines with physiological cytokine concentrations. a** Representative plots of HEK293T cells infected using lenti viral vectors with RFP670 under the control of the G1K06H1 promotor and ZsGreen controlled by the ef1α core promotor. At 72 h following infection, the cells were treated for 48 h with 500 U/ml indicated cytokines and for 18 h under hypoxic or normoxic conditions, harvested and analyzed by flow cytometry. Data shown are ZsGreen-positive, single-discriminated and DAPI-negative results. **b** From the top panel to the bottom—cells were treated for 48 h with 500, 125, 32, 8, 2 or 0 U/mL of TNF respectively and veering IFN concentrations and placed under hypoxic or normoxic conditions for 18 h, cells were analyzed by flow cytometry as described above. Showing average of duplicates, error bars indicate standard deviation. **c** From the top panel to the bottom—cells were treated for 48 h with 500, 125, 32, 8, 2 or 0 U/mL of IFN respectively and veering TNF concentrations and placed under hypoxic or normoxic conditions for 18 h, cells were analyzed by flow cytometry as described above. Showing average of duplicates, error bars indicate standard deviation. **d** Ratios of reporter expression with different cytokine concentrations under hypoxic or normoxic conditions. Results are from one representative experiment of four performed.

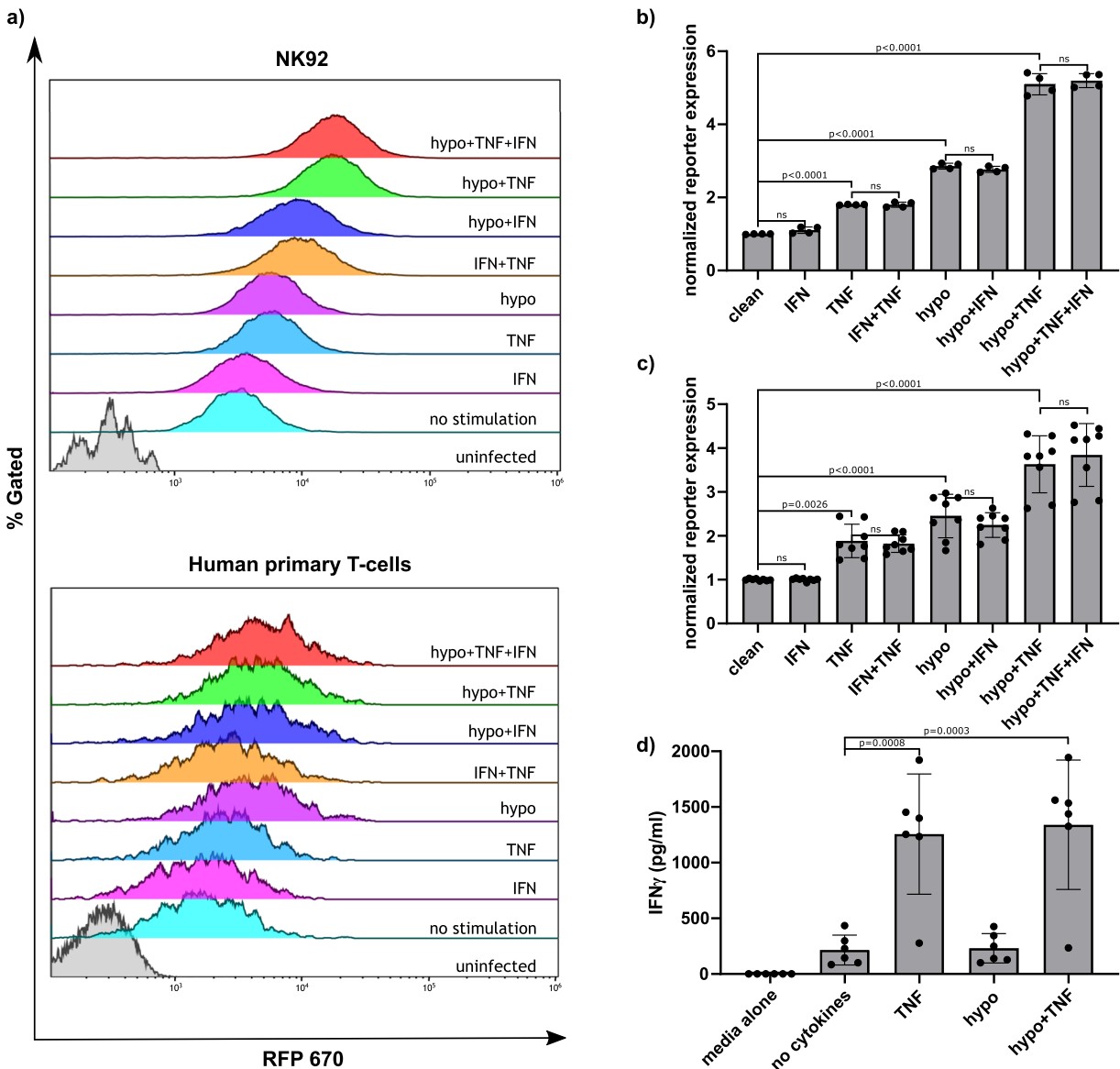

**Fig. 5 CARTIV promotors show an additive response only to TNFα and hypoxia in primary human T cells and NK92 cells due to autocrine IFNγ secretion. a** Representative plots of primary human T cells or NK92 cells infected using lenti viral vectors with RFP670 under the control of the indicated CARTIV promotor and ZsGreen controlled by the ef1α core promotor. At least 72 h following infection, the cells were incubated for 48 h with the indicated cytokines at 250 U/mL and for 18 h under hypoxic or normoxic conditions, harvested and analyzed by flow cytometry. Data shown are ZsGreen-positive, single-discriminated, and DAPI-negative results. **b**, **c** Geometric mean of RFP670 in ZsGreen-positive cells normalized to unstimulated cells, showing the average of duplicates, an ordinary one-way ANOVA was performed to test for significance, error bars indicate standard deviation of NK92 or human primary T-cells. **d** Supernatants of primary human T cells incubated for 48 h with 250 U/mL indicated cytokines and for 18 h under hypoxic or normoxic condition were collected, and ELISA for IFNγ was performed. Showing average of triplicates, an ordinary one-way ANOVA was performed to test for significance, error bars indicate standard deviation. Results are from one representative experiment of four performed.

revealed endogenous secretion (Fig. 5c, showed for primary human T cells), suggesting for possible positive auto-regulation in such culture conditions.

**CARTIV G1K06H1 can induce the expression of a functional CAR.** Following the good induction of reporter-genes, we next aimed to find if CARTIV promotor could also induce CAR receptors to the cell's surface. Therefore, we constructed a Herceptin-based 3rd generation CAR (Supplementary Fig. 1) Open Reading Frame (ORF) just following the G1K06H1 promotor to study its direct induction by stimuli. The CAR employed was composed of a Herceptin based scFv, a CD28 and 4-1BB costimulatory motifs and a CD3ζ chain[38] (Fig. 6a), suggesting for activation of effector cells when encountering target cells that

express high levels of the cognate ERBB2[38]. To assess for surface-expression, cells were stained using an ERBB2-Fc chimeric protein, the CAR cognate ligand. ZsGreen+ positive T cells manifested clear expression of the CAR following the stimulation, as compared to non-stimulated ZsGreen+ positive T cells, and to the ZsGreen− T cells in the same well (Fig. 6b). We then tested the functionality of this CAR by employing target cells that manifest high, dull or no expression of the ERBB2 target protein on their cell membrane (Supplementary Fig. 6a, b). Activity of the T cells was measured using the CD107a-based degranulation assay. The percentage of degranulating cells was calculated as the fraction of CD107a positive cells derived from the ZsGreen positive population. Following stimulation, the T cells incubated with ERBB2dull and ERBB2high target cells showed substantial levels of

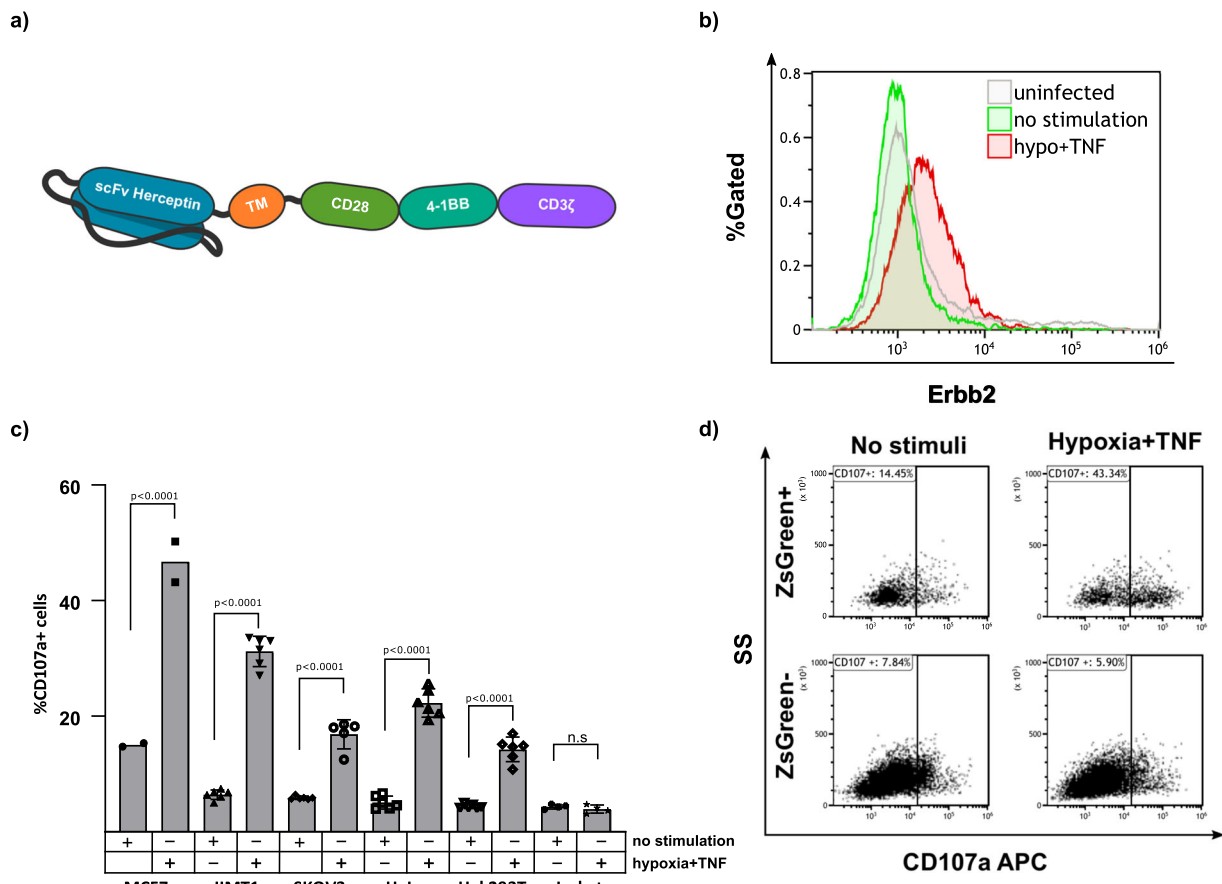

**Fig. 6 CARTIV promotors allow inducible expression of a Herceptin-based CAR, leading to degranulation against HER2-expressing target cells.**
**a** Schema of the CAR used. **b** Representative plot of human primary T cells incubated for 48 h with 250 U/mL of TNFα and for 18 h under hypoxic or normoxic conditions. The cells were then stained using an Erbb2-Fc chimeric protein and analyzed by FACS. Data shown are ZsGreen-positive, single-discriminated, and DAPI-negative results. **c** %CD107a positive and ZsGreen positive human primary T cell following 48 h with 250 U/mL TNFα and for 18 h under hypoxic conditions and incubation with the indicated target cell line. Percentage of CD107a positive cells is normalized to the total number of ZsGreen positive cells. An ordinary one-way anova was performed to test for significance, Error bars indicate standard deviation. **d** CD107a expression following incubation with MCF7 cells of ZsGreen positive and negative human primary T cells with and without the indicated stimuli. Results for **b** are from one representative experiment of four performed; **c** summarizes 2–6 experiments (target-dependent).

degranulation when compared to HER2 negative Jurkat cells, with different basal-levels differences among target-cells probably due to secreted cytokines (Fig. 6c). The degranulation observed for the T cells induced to express the CAR and incubated with HER2 positive cells was substantially higher (a 3.1, 4.82, 3.02, 4.14, and 2.58 fold change for MCF7, JIMT1, SKOV3, HeLa, and HEK293T target cells accordingly when compared to the non-stimulated T-cells) than the degranulation observed for HER2 negative target Jurkat cells. Therefore, the CAR transcribed following the stimuli of the CARTIV promoter and expressed on the cell's surface is also functional specifically against its cognate target. Figure 6d show experimental raw data for MCF7 as target cells. CAR-positive T cells (ZsGreen$^+$ cells) are substantially activated upon stimuli, when incubated with HER2-positive target cells (Fig. 6d, upper panels), while CAR-negative T cells (ZsGreen$^-$) are not activated upon stimuli on HER2-positive target cells (Fig. 6d, bottom panels). These data support the possible improvement of CAR-T activity against tumor, but not against normal tissues.

**G1K06H1 promoter in NK92 cells is responsive to a tumor microenvironment when compared to a non-tumor site in a CDX model.** Next, we investigated whether CARTIV promoter could be induced in vivo within the TME as compared to a non-TME site. A cell line derived xenograft (CDX) model was

established by injecting the HER2-positive JIMT-1 tumor cells[39] into immunocompromised NSG mice. When JIMT-1 tumors reached ~150 mm³, NK-92 cells harboring RFP670-encoded by the G1K06H1 promoter were injected intra-tumorally and in parallel (into the same mouse) subcutaneously in Matrigel (Fig. 7a). Forty-eight hours after inoculation, mice were sacrificed and tumor and matrigel were extracted and dissociated. NK-92 cells, residing within the tumor, Matrigel were gated based their on GFP expression (Supplementary Fig. 7A), and their expression of the RFP670 reporter was measured by flow cytometry. When we compared the RFP670 reporter expression in NK-92 from the tumor site to NK-92 from the Matrigel, a clear enhancement of expression was observed in the tumor site (Fig. 7b, representative). The increase of expression at the tumor site was statistically significant (Fig. 7c). In additional set of experiments on mice bearing JIMT-1 CDXs, NK-92 cells harboring RFP670-encoded by the G1K06H1 promoter were injected intra-tumorally and in parallel (into the same mouse) intravenously. Forty-eight hours after inoculation, we analyzed RFP670 expression in GFP$^+$ cells derived from the TME or from the blood. Again, a clear significant enhancement of RFP670 expression, a 1.57 fold increase, was observed in cells derived from the tumor site as compared to cells from a non-TME source, i.e. the blood (Supplementary Fig. 7B). These results suggest that the G1K06H1 CARTIV

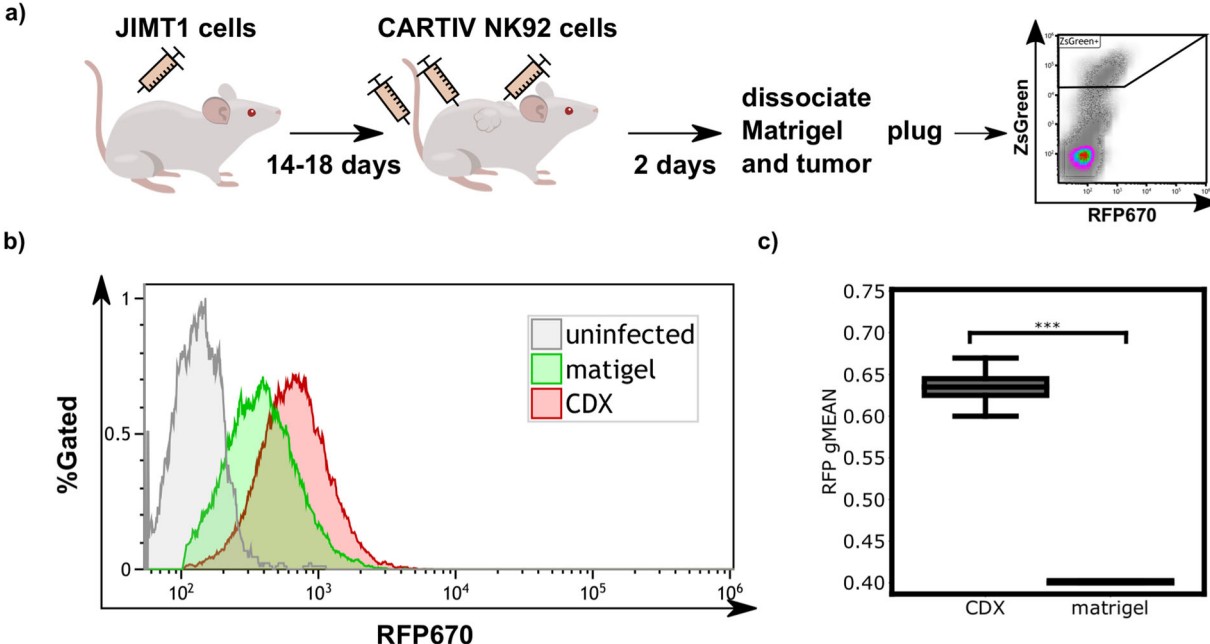

**Fig. 7 The G1K06H1 CARTIV promotor is induced in the tumor microenvironment. a** Schema of the experimental setup used to test the G1K06H1 promotor in vivo. **b** Representative FACS plot of the RFP670 reporter of ZsGreen positive cells injected subcutaneous or intratumoral. Data are single-discriminated and DAPI-negative results. **c** Summary of reporter expression in the CDX compared with the Matrigel plug in 6 mice; exclusion criterion was less than 500 cells detected. Number of tested tumors is eight; a *t*-test was performed, box borders represent upper and lower quartiles. Solid lines represent the median and dashed line the mean. Whiskers represent maximum and minimum. *P-value < 0.05; **P-value < 0.005; ***P-value < 0.0005.

promoter is induced in vivo in response to the tumor micro-environment when compared to a non-tumor site.

## Discussion

CAR-T gained clinical success, with approved treatments against uncured cancers[40], and hundreds of current clinical trials[41]. However, the risk of harming normal tissues is preventing its application to many patients[42]. Focusing effector immune cell's activities onto tumors, while sparing normal healthy tissues can work together with current CAR-T approach, as well as with any other adoptive-transferred engineered immune cells. The CAR-TIV approach presented hereby is suggesting the combination of multiple promoter-elements that create an effective response to numerous cytokines and environmental conditions that are prevalent within TME, and may bring improved immune-focus ON-targets with lower to no OFF-target damage.

The potency of T cells to identify specific target cells and destruct them caught the imagination of scientists for decades[43,44]. It took many years and innovations to realize basic ideas into clinical practice, and have FDA-approval of CAR-T. Currently, the main limitation for new CAR-T that can eradicate additional types of tumors is safety, including cytokine release syndrome[45], and destruction of ON-target OFF-tumor cells[2]. If the specified tumor antigen is expressed on normal cells, even at low levels, CAR-T might recognize and attack the tissue, causing potential lethal damage[2]. Multiple approaches are being developed to increase CAR-T safety, including: ablation of the T cell[46]; two-step activation[47]; small-molecule ON-switch[48]; soluble-receptors[49]; dual-specificity with dissociated signaling[11]; or TKI-switch[50] to name a few. Many studies aimed to further design specific features in cytotoxic cells. For example, Kulemzin and Gorchakov smartly introduced tandem-repeats of NFAT-response elements, or of *NFkB*-response elements into NK-cells or T cells, gaining inducible transgene expression following engagement by target-cells[51]. Such design can facilitate additional

features by CAR-T cells, but it is still specific for the single tumor-antigen. CARTIV approach includes multiple PREs having specificity to TME rather than a tumor-antigen, in order to reduce risk of ON-target OFF-tumor activities.

Turning CAR-T ON/OFF throughout the body may allow to reduce cytotoxic damage, but also alleviate the anti-cancer activities. The CARTIV approach brings the ability to focus CAR-T, or any engineered immune-cell, onto spatiotemporal zone within the body. This may allow to reduce or even completely avoid the ON-target OFF-tumor activity while sustaining high effector function on the TME.

Inducing expression of effector-genes to a limited spatio-temporal resolution is an inherent feature of the immune system. Both innate and adaptive cells change gene expression following activation[52]. The CARTIV promoter includes response elements for some of the main inflammatory cytokines that define in vivo the zone of effector cells destructive activities. Although, IFNγ is usually referred to as an antitumor cytokine it can also mediate pro tumorigenic transformation and progression. Similarly, TNFα is secreted by inflammatory cells and is related to inflammation-associated carcinogenesis. It has been previously shown that the presence of IFNγ in the TME could be directly associated with tumor virulence. Our data is presenting the potent ability to combine promoter elements of various response factors and gain an induced activation by the combination of external stimulants. The length of the CPRE we used in the CARTIV approach is surprisingly short. While endogenous promoters are usually including proximal promoters that are around hundred bases before and after the transcription Start Site (TSS), and enhancer elements that may reside thousands and even million-base, we find sufficient abilities of short 200 base pair promoters to include multiple PREs. This is perhaps not surprising since the actual binding-sites of transcription factors are usually less than dozen bases. We may speculate that natural promoters are spread over larger length simply since there is no evolutionary reason for

them to squeeze. Using rational-design and prior knowledge is suggesting for the opportunity to create synthetic promoters with complex response to multiple factors within relatively short sequences of DNA.

Special note should be given to the non-trivial combination of the CPRE location that we observed in the CARTIV approach. Combinatorial response of CARTIV promoters is designed to allow robust focus of activity within TME, and little or even no activity in normal tissues. Considering safety, having single-factor regulation will possibly improve the constitutive-expression which is currently used, but will inevitably have limited abilities to restrict activity within TME only. Synergistic induction is therefore desired to gain specificity into the tumor microenvironment. Surprisingly, the combination of multiple CPREs into a concise CARTIV promoter found non-trivial significance for the exact sequences and for the order by which the different CPREs are located (Fig. 1). Especially surprising was the dramatic effect of the hypoxia CPRE localization (Fig. 3).

In the CARTIV design we aimed to achieve synergistic induction by multiple factors. The factors used in this study may have a substantial range that in turn may affect the response elements activities. TNFα is a major inflammatory cytokine, having systemic and local activities; IFNγ is having a more local-microenvironment function that was reported specifically to affect cytotoxic T cells via local paracrine and even autocrine fashion[53]; hypoxia is considered a hallmark of cancer, but levels may range from pO2 of 80 mmHg down to zero, with substantial heterogeneity within tumor microenvironment[54]. Interestingly, IFNγ is also secreted from T cells and NK cells, possibly making a positive feed-forward activation, but also possibly limiting usage in case one require the lowest background expression. The later may require further modification of our technology, for example, by replacing the IFNγ CPRE.

As mentioned above, the range of biological-activity in vivo is known to differ substantially, as TNFα is having the long-range diffusion and IFNα is local, and hypoxia is simply doing as the pO2 for each cell. Our data indicate that the synergistic effect on CARTIV promoters is largely dependent on the concentrations of the relevant factors (Fig. 4). Strikingly, the synergism of TNFα + IFNγ + hypoxia was most prominent at the low-range of cytokine concentration tested, which correspond with physiological levels[34,35]. This is most likely a feature of the specific CARTIV promoter in these experiments, as other vectors having multiple binding-sites to the same factor showed higher response to same cytokine levels (Fig. 2). Therefore, we may suggest that synthetic promoters can be designed and optimized to the relevant concentrations of factors; the synergism of multiple factors is most beneficial when each is contributing some mediocre induction. Future studies may use these principles to further design promoters for additional factors and gaining sensitivities to their relevant physiological levels. It is probable that by combining different or additional CPREs, employing various ratios between them and testing alternative relative CPREs locations, one could achieve a more robust promoter activation and CAR expression that will then be assessed in vivo for successful tumor regression followed by employing this approach in a clinical setting.

To summarize, CARTIV promoters are an approach to focus engineered immune cells activities into tumor microenvironment. With growing interest in CAR-T, and their safety limitations, we have major interest in applying CARTIV for CAR-T. Further use of such CARTIV promoter may provide immune-focus for any engineered adoptive-transfer approach of effector cells. We anticipate multiple utilization for the synthetic promoters combining multiple response elements.

## Methods

**Cloning.** Synthetic promoters were designed based on curated data and combinations of binding-sites for TME-induced transcription factors, together with specified linkers. DNA was synthesized and cloned in shuttle vector (HyLabs, Rehovot, Israel). CARTIV promoters were amplified with primers containing restriction-enzyme sites using PrimeSTAR Max DNA Polymerase (Takara, CA, USA). DNA fragments were resolved on agarose gel, extracted, digested, and cloned into pHAGE2 plasmids using NEB enzymes.

**Tissue culture.** HEK293T and JIMT-1 cells were grown in DMEM (Gibco, MA, USA) containing 10% serum, pen-strep, HEPES, L-glutamine, non-essential amino acids, and sodium pyruvate (all from Biological Industries, Beit Haemek, Israel). NK-92 cell were grown α-mem (Gibco, MA, USA) containing 10%FBS, 10% horse serum, pen-strep, HEPES, L-glutamine, non-essential amino acids and sodium pyruvate (all from Biological Industries), Myo inositol (Sigma, MO, USA), folic acid and β-mercapto-ethanol (Gibco, MA, USA). All tissue cultured cells were kept in a humidified 5% $CO_2$ incubator.

**Cytokines concentrations and handling.** Human recombinant IFNγ and TNFα were purchased from PeproTech (Rehovot, Israel); cytokines were reconstituted and stored according to the manufacturer recommendations. Both IFNγ and TNFα had a specific activity corresponding to $2 \times 10^7$ Units/mg.

**NF-κB reporter.** HEK293T cells plated at $1 \times 10^6$ per well in a 6-well plate and the following day were co-transfected with N1-GFP and a NF-κB-mCherry reporter plasmid (pNFκB-MetLuc2-Reporter modified into pNFκB-mCherry by conventional cloning) using JetPrime® reagent (Polyplus, Illkirch, France) according to the manufacturer recommendations. To following day calls were plated again at $5 \times 10^6$ per well in 24-well plate and added TNFα to final concentration of 500–7 U/mL for 24 and 48 h and kept in a humidified 5% $CO_2$ incubator. Cells were harvested and washed in PBS 2%FCS and suspended with DAPI 1 µg/mL and FACS measured using Beckman Coulter® Gallios™ flow cytometer. Data were analyzed using Kaluza™.

**MHC class 1 staining.** HEK293T cells plated at $1 \times 10^5$ per well in a 96 well Flat bottom and incubated with IFNγ to final concentrations of 250–0.5 U/mL for 24 and 48 h and kept in a humidified 5% $CO_2$ incubator. Cells were harvested and washed in PBS 2%FCS and stained with anti HLA-A,B,C—PE (clone w6/32; BioLegend, CA, USA) as suggested by the manufacturer. Cells were washed once in PBS 2%FCS, suspended with DAPI 1 µg/mL and FACS measured using Beckman Coulter® Gallios™ flow cytometer. Data were analyzed using Kaluza™.

**Human primary T cells culture.** Primary T cells obtained from healthy consenting donors, 10 mL of blood was taken using a DG-veinset (VSET21) into LH Lithium Heparin tube (Greiner Bio-One, Austria, Kremsmünster). Blood was diluted in a 1:1 ration using PBS 2% FBS and loaded to a Ficoll (Mp-Bio, USA, CA) and separated by centrifugation at $400 \times g$ for 30 min. Mononuclear cells were collected as the interphase, washed twice using PBS 2% FBS, counted and plated at $3.75 \times 10^6$/mL in 24-well plates in RPMI 10% human serum (Sigma, MO, USA), 300 U/mL rhIL2 (PeproTech) and 50 ng/mL of anti-human CD3 (BioLegend). After 48 h cells were harvested and re-plated in complete media without OKT3. Cells were kept in a humidified 5% $CO_2$ incubator.

**Lenti virus production and concentration.** HEK293T cells were grown to ~90% confluence in 10 cm plates in DMEM containing 10% serum, pen-strep, L-glutamine, HEPES, non-essential amino acids and sodium pyruvate (all from Biological Industries). We transfected 10 µg of pHAGE2 lent vector and 3 µg of packaging plasmids tat, rev, hgpm2 and vsvg in a 1:1:1:2 ratio, using JetPrime® reagent (Polyplus). We changed the medium on day 1, and collected LVs by changing media on day 2, 3, and 4. The media containing LVs was filtered through 0.45 µ PVDF membrane into ultra-centrifuge tubes (Beckman Coulter), and centrifuged for 90 min at 17,000 RPM 4 °C. The LV's pellets were suspended and used at the same day or aliquoted and kept in −80 °C.

**Human primary T cells infection.** LVs were spin loaded to 24-well non-treated plates (Thermo Fisher) that were coated with 60 µg/mL retronectin (Takara) at 4 °C overnight. Primary T cells that were activated for 48 h, as described above, and added with fresh media supplemented with 10% human serum and 300 IU hrIL2 at $1.25 \times 10^5$–$2 \times 10^5$ cells/mL/well. Plates were spun at $500 \times g$ at 32 °C for 10 min and returned to incubator. The next day cells were suspended, transferred to a new 24-well plate, and cultured further for up to 21 days.

**Hypoxic conditions.** Hypoxic chamber was used with gas mixture of 5% $CO_2$ 0.3% $O_2$ and 94.7% $N_2$ at 20 L/min for 3–5 min and then sealed and placed at 37 °C for 16–20 h before analysis.

**CARTIV promoter activity assay**. HEK293T cells plated at $1 \times 10^5$ per well in a 96 well Flat bottom; NK92 or human primary T cells at $2 \times 10^5$ in a 96 well U-shape. Cytokines IFNγ, TNFα (PeproTech) added to final concentrations of 32–500 U/mL as indicated per experiment. In experiments involving Hypoxia, it was induced for the last 16–20 h of the experiment. Cells were harvested, washed once in PBS 2% FCS, suspended with DAPI 1 μg/mL and FACS measured using Beckman Coulter® Gallios™ flow cytometer. Data were analyzed using Kaluza™.

**CARTIV promoter kinetics assay**. HEK293T cells plated at $5 \times 10^5$ per well in a 24-well plate; cells were either stimulated using IFNγ and TNFα 500 U/mL for 48 h and then washed twice in complete DMEM or supplemented with IFNγ and TNFα 500 U/mL before starting the measurement. Cells were imaged in a Lionheart™ FX Automated Microscope every hour for 48 h. Data were filtered by gating on GFP positive cells and then was plotted using a python script utilizing the matplot and seaborn libraries. Growth and decay curves were fitted to data using the curve_fit function in the scipy library.

**CAR expression assay**. Following activation, human primary T cells were washed once, stained using an ERBB2-Fc chimeric protein (R&D, USA, MN) 4 μg/ml for 1 h on ice, washed twice in PBS 2% FCS and stained with a Goat anti human APC IgG (The Jackson Laboratories, USA, ME). FACS measured using Beckman Coulter® Gallios™ flow cytometer. Data were analyzed using Kaluza™.

**T cell functional assay**. Primary human T cells were plated at $2 \times 10^6$/mL in 24-well plate, and induced as indicated above. The effector T cells were labeled using DiI (Thermo Fisher), target cells were mixed at various ratios with effector cells. Degranulation was assayed using anti human CD107a APC (Biogems, Israel, Rehovot). Cells were incubated at 37 °C 5% $CO_2$ for 4 h and analyzed by FACS as above.

**ELISA**. Primary human T cells or NK92 cells were plated at $5 \times 10^5$ per well in 96 plates and supplemented with 250 UI/ml of IFNγ, TNFα or both, after 24 h cells were placed at hypoxic conditions for 16–18 h. Supernatants were collected cytokines quantified using standard ELISA as described elsewhere[55].

**In vivo CARTIV activation assay**. NOD.Cg-Prkdc Il2rg/SzJ (NSG) mice were purchased from The Jackson Laboratory. For JIMT-1[39] cell line-derived xenografts, 6-week-old female mice were injected s.c in the flank with $5 \times 10^6$ cells in 100 μl of matrigel. Two weeks later, when the tumors reached ~150 mm³, tumor-bearing mice were injected again with $5 \times 10^6$ NK92 CARTIV+ cells i.t, i.v or s.c in matrigel in the neck. Two days later, tumors were dissociated for a single cell suspension using the gentleMACS Dissociator. For blood samples RBCs were lysed using ACK solution for 5 min at RT and washed using PBS. Samples were 70 μm strained, washed twice using PBS and suspended in PBS 2%FCS with DAPI 1 μg/mL and FACS measured using Beckman Coulter® Gallios™ flow cytometer. Data were analyzed using Kaluza™. Box plot was plotted using a python script utilizing the seaborn library. A two-tailed $t$ test was performed using the scipy library. All animal experiments were carried out under the Institutional Animal Care and Use Committee (IACUC) of Ben-Gurion University of the Negev (BGU's IACUC) according to specified protocols aimed to ensure animal welfare and to reduce suffering. The animal ethical clearance protocol numbers used for this study are IL-80-12-2015 and IL-29-05-2018.

**Statistics and reproducibility**. For FACS data shown are means ± s.d. of geometric mean of the indicated gate from three independent experiments unless otherwise noted. A two tiled $t$ test was performed to compere between two groups. Unless otherwise noted, all graphs were generated using GraphPad as well as the statistical analysis. All data analyzed in this study is available on request from the authors.

## Data availability

Supplementary Data 1 gives all data on all experimental replicates presented in the main work. Additional Source data regarding figures and Supplemental Figures will are available on request from the corresponding author.

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

## Acknowledgements

The first author would like to thank Dr. Idit Shub, Gal Tsachor and Ilan Edelson for there long lasting support and endless belief. This work was supported by the Israel Science Foundation grant 2484/19 (A.P.), the US/Israel Binational Science Foundation grant 2019337 (A.P.), the joint NRF (Singapore)-ISF grant 3127/19 (A.P.), the Ministry of Health grant 3/15080 (R.G. and A.P.), the NIBN-CARTIV grant. and the DKFZ-MOST 3-14370. The funders had no role in study design, data collection and analysis, decision to publish, or preparation of the manuscript.

## Author contributions

Y.G. performed a large portion of the experimental work, analysis, and contributed substantially to the design and data interpretation. O.S. performed substantial experimental work. A.O. contributed to data analysis in the time laps experiments and performed graphical editing to the figures. A.C. performed experimental work in our proof of concept stage. K.K. and K.Y. established and maintained our animal model under the supervision and guidance of M.E. R.G. contributed to the study design, supervision, and data interpretation. A.P. led the study, supervised the experimental work and data analysis, and let the writing and styling of the study.

## Competing interests

The authors declare no competing interests.
