## [Peer Review File · Communications Biology]

Reviewers' comments:

Reviewer #1 (Remarks to the Author):

In this article, Greenspan et al explore the use of synthetic promoters based on IFN γ , TNF α and hypoxia responsive elements to control gene expression. The authors describe this technology in the context of chimeric antigen expression (CAR) and propose that this method could circumvent off-target toxic effects via specific expression of the CAR in the tumor microenvironment. While being an exciting idea, the concept doesn't seem to be completely novel (e.g. BMC Medical Genomics volume 12, Article number: 44 (2019)). However, if thoroughly explored and highlighting the benefits over previously published systems, the article would be of interest for the community and the broader audience. As it is, the amount of data offered is not sufficient to convince that this system will confer advantages to CAR T-cell activity in the tumor microenvironment, since there is a very limited amount of data generated with CAR T-cells.

It would be useful if the authors could clarify the rationale to choose IFN γ , TNF α and hypoxia, specially when in an immunosuppressed tumor environment, the presence of immune-related cytokines could be low and not sufficient to activate the expression of the CAR.

Is there a reason why the authors do not test individually GCPRE and KCPRE? What is the reasoning to use the 60% KCPRE? Has this been described before? Have they tested different sizes for this element?

While the location of the hypoxia responsive elements is assessed and shown to be relevant to the function of the element, for GCPRE and KCPRE no location effects are explored. It would be interesting to know if different PRE locations confer an increase in expression or, at least, the rationale for not testing different configurations. Also, additional controls such as no responsive elements and a constitutive promoter would be desired.

The authors refer to the putative function of HRE downstream of the TSS- did they test this? Exploring these elements would add robustness to the data.

Figure 4B- it appears as the addition of Hypoxia to IFN γ and TNF α , does not confer a benefit.

Figure 4C is lacking non stimulated control (normoxic vs hypoxic) and it is not possible to assess the level of RFP670 expression over basal levels. Also data on IFN γ stimulation alone is missing.

Can the authors clarify the levels of IFN γ and TNF α expected to be found in tumors? Even if the synergistic effect with hypoxia is best at 32IU/mL, the expression is still lower than in other conditions- this leads to question whether at this level of expression will be functionally relevant in the tumor context- can the authors comment on this? This is a main point of concern overall.

Addition of IFN elements seems to be dispensable to both TNF α and hypoxia in T- and NK- cells. The absence of increased expression in T-cells and NK92 cells via addition of IFN γ should be explored by testing constructs without the GCPRE elements. After all, T- and NK cells are a more relevant model than HEK293T cells. Have the authors explored the possibility of testing other PREs that could potentially enhance expression in T- and NK- cells?

It is not clear where ZsGreen comes from in CAR transduced cells for Figure 6- it would be useful if the authors could clarify this. Also degranulation levels are not really convincing. For proof of principle, the authors should at least perform cytotoxicity assays and cytokine releasing assay in order to accurately measure CAR activity and compare this data to a constitutively expressed CAR.

Authors should further test the activity of the CAR in vivo in order to assess the ultimate functionality of this technology and consolidate the proof of concept. It is again unclear which construct was injected into the mice- did it have the CAR or was it only the RFP670 expressing construct? In any case the expression of RFP670 in Figure 7B is pretty low and it is not possible to determine if this level of expression is functionally significant and will have a biological effect. Thus, the authors should also test CAR T-cell activity in vivo for tumor clearance and show that this activity is tumor specific and not elicited by normal tissues.

Did the authors examine other parts of the mouse for activation of CAR? Since the authors present this technology as an approach to overcome off-target toxicity, this should be more thoroughly investigated.

Reviewer #2 (Remarks to the Author):

This manuscript presents a series of vectors in which transgene expression is modulated by exposure to inflammatory cytokines and hypoxia. While conceptually of interest, I have several major reservations about the work.

1. Authors should justify why they selected to make gene expression responsive to IFN γ and TNF α , rather than tumour-associated inhibitory cytokines such as TGF β for example. What levels of these pro-inflammatory cytokines are actually present in typical solid tumors and how does that relate to concentrations of these cytokines used in experiments shown.
2. In contrast to the cytokines chosen, the logic of rendering gene expression inducible in hypoxia is clear, given its prevalence in advanced solid tumors. However, authors should acknowledge others who have attempted something similar previously eg Juillerat et al Sci Rep (2017) 7: 39833.
3. Expts in Fig 2 entail the use of a single concentration of cytokine without justification. Dose response and time response curves should be shown in these experiments (e.g. how long to switch on; how long to switch off). Similarly, the performance of the hypoxia-responsive promoter should be evaluated across a spectrum of oxygen tensions and over time.
4. The number of replicate expts and nature of the error bar should be indicated in the legend to all relevant figures.
5. The performance of the mini-TK promoter alone should be demonstrated throughout the manuscript. This control is also required to assess leakiness of the other promoters when tested in the absence of the corresponding stimulus (e.g. cytokines and/or hypoxia).
6. The experiment shown in Figure 4 seems to be a single example and lacks many controls.
7. Background production of cytokines such as IFN γ by activated T-cells or NK cells is a clear drawback of the proposed strategy.
8. Reproducibility of data in Fig. 6 is unclear.
9. Therapeutic activity of CAR T-cells is not shown in vivo.

Minor

1. Describe (tumor?) origin of JIMT1 cells.
2. Provide nucleic acid sequences of all CARs in supplementary material.
3. Supplementary Figure 1A does not show expression of IFN γ and TNF α receptors by HEK293T cells.

Reviewers' comments:

The line numbering we are referring to in the text are in **simple markup**

Reviewer #1 (Remarks to the Author):

In this article, Greensphan et al explore the use of synthetic promoters based on IFN γ , TNF alpha and hypoxia responsive elements to control gene expression. The authors describe this technology in the context of chimeric antigen expression (CAR) and propose that this method could circumvent off-target toxic effects via specific expression of the CAR in the tumor microenvironment. While being an exciting idea, the concept doesn't seem to be completely novel (e.g. BMC Medical Genomics volume 12, Article number: 44 (2019)). However, if thoroughly explored and highlighting the benefits over previously published systems, the article would be of interest for the community and the broader audience. As it is, the amount of data offered is not sufficient to convince that this system will confer advantages to CAR T-cell activity in the tumor microenvironment, since there is a very limited amount of data generated with CAR T-cells.

Thank you for describing the study as a thorough exploration of the new suggested concept over previously published systems. We thank the reviewer for relating to the novelty of the original idea and pointing the reference above. We discuss the differences between our idea and the ref above in the text (lines 368-374 in the text ref number 55). In short, the study of Kulemzin and Gorchakov had smartly engineered either NFAT-response elements, or NF κ B-resonse elements, or portions of the CD69-promoter. This already gained good induction by cytotoxic cells after challenge with specific target-cells. CARTIV approach brings multiple promoter-response-elements, and their combinations. We demonstrate the induction by TME factors, rather than activation through the single chimeric receptor, with the aim to gain confined responce against tumor-target and limit the ON-target OFF-tumor risk.

It would be useful if the authors could clarify the rationale to choose IFN γ , TNFalpha and hypoxia, especially when in an immunosuppressed tumor environment, the presence of immune-related cytokines could be low and not sufficient to activate the expression of the CAR.

It is an important point. Thank you. In the submitted MS, we shortly discuss this issue and cite four references. We now expanded our discussion and added five additional references (numbered 16-20 in the Ms).

Although IFN γ is often represented as antitumor cytokine, IFN γ can efficiently mediate signaling that elicits pro-tumorigenic transformations and promotes tumor progression. Similarly, as a pro-inflammatory cytokine, TNF α is secreted by inflammatory cells, which could be involved in inflammation-associated carcinogenesis. Others and we have previously showed that the presence of IFN γ in the TME could be directly associated with tumor virulence (line 84 ref 16 in the revised Ms). We added 3 additional references (including ours) to better explain our decision to begin our initial studies with IFN γ and TNF α . We also strengthened the discussion for our decision to add hypoxia element, which is a recognized driver in the TME of tumor metastasis line 84 ref 17-20 in the revised Ms). In accordance, we expanded the text in the discussion (lines 384-388).

Is there a reason why the authors do not test individually GCPRE and KCPRE? What is the reasoning to use the 60% KCPRE? Has this been described before? Have they tested different sizes for this element?

We tested the individual GCPRE and KCPRE - results are shown in the revised Ms in supplemental Figure 3 and described in the text in lines 216-231. Both have effect; KCPRE is better inducer than GCPRE; thus, we compared one KCPRE to 2, 4, and 6 GCPRE (supplemental Figure 3). Our goal was to achieve synergism between multiple promoter elements activated by conditions characterizing the TME. Since KCPRE alone was quite potent even as compared to 6 GCPRE, we decided to study a fraction of the KCPRE combined with GCPRE and HCPRE. On Figure 2, one can observe that indeed the best synergism is achieved with G1K0.6 as compared to G1K1, G2K2, and G3K3.

While the location of the hypoxia responsive elements is assessed and shown to be relevant to the function of the element, for GCPRE and KCPRE no location effects are explored. It would be interesting to know if different PRE locations confer an increase in expression or, at least, the rationale for not testing different configurations. Also, additional controls such as no responsive elements and a constitutive promoter would be desired. The authors refer to the putative function of HRE downstream of the TSS- did they test this? Exploring these elements would add robustness to the data.

(i) We agree with the reviewer that it would be interesting to know if different PRE locations confer an increase in expression. Yet, investigating a full-factorial matrix of the CARTIV promoter possible compositions for a 3-PRE promoter including multiple relative locations of the PREs is a tedious task that will require a significantly more time and efforts, beyond the scope of the current report on this technology.

(ii) Thank you for pointing this. We added these controls (no responsive element & constitutive promoter) in the revised supplemental Figure 2D and in the text (lines 223-225).

Figure 4B- it appears as the addition of Hypoxia to IFN γ and TNF α , does not confer a benefit.

The revised Figure 4 extensively details the effect of the Hypoxia addition on the two cytokines. Indeed, when studying the hypoxia element in high levels of IFN γ and TNF α , the effect of adding the hypoxia is less dominant (revised Fig. 4A and lines 273-276) but as detailed in the new Figure 4C, hypoxia element significantly add to the efficiency of the triple-CPRE-promoter in lower levels of cytokines. These are physiological levels as described in the answer to the comment below.

Figure 4C is lacking non-stimulated control (normoxic vs hypoxic) and it is not possible to assess the level of RFP670 expression over basal levels. Also data on IFN γ stimulation alone is missing.

Figure 4 is now completely revised. Additional experiments were performed to fully characterize the various combinations of cytokine concentrations. Non-stimulated control (normoxic vs hypoxic) in various combination of cytokine levels, including no cytokine-

stimulation at all, is shown in the zero axes-interception point in each of the 12 insets in Figure 4C. IFN γ stimulation alone, over range of cytokines, appears now in the normoxic graph in inset in which TNF α is zero units. Revisions are described in the text (revised Fig. 4A and lines 271-276).

Can the authors clarify the levels of IFN γ and TNF α expected to be found in tumors? Even if the synergistic effect with hypoxia is best at 32IU/mL, the expression is still lower than in other conditions- this leads to question whether at this level of expression will be functionally relevant in the tumor context- can the authors comment on this? This is a main point of concern overall.

32 IU/ml equals to 2 ng/ml (revised methods for the conversion line 110-113) which better reflect the functional levels of these cytokines in the body (new references 41 and 20) and in the text lines 272-281 in results and methods showing the conversion between weight to units). In short, levels of either IFN γ or TNF α in which hypoxia is significantly contributing for our 3CPRE promoter, is ~2 ng/ml; several reports in the literature, locate the observed physiological levels of TNF α and IFN γ to this range (new refs 41 and 20 and in the text 422-427). It is also important to consider that the TME is an heterogeneous environment and in hypoxic areas cytokines levels could be lower and vice versa. Therefore, the 3-CPRE synthetic promoter could be the answer also in these circumstances, keeping certain levels of activity throughout the heterogeneous TME.

Addition of IFN elements seems to be dispensable to both TNF α and hypoxia in T- and NK- cells. The absence of increased expression in T-cells and NK92 cells via addition of IFN γ should be explored by testing constructs without the GCPRE elements. After all, T- and NK cells are a more relevant model than HEK293T cells. Have the authors explored the possibility of testing other PREs that could potentially enhance expression in T- and NK-cells?

Thank you for this important comment. We are definitely exploring now new 3CPRE promoters having promoter elements induced by other cytokines characterizing the tumor microenvironment (e.g TGF β instead of IFN γ). Yet, there is some importance for the IFN γ GCPRE in the context of T and NK cells. The T/NK cells grown *in vitro* auto-saturate the GCPRE element by autocrine secretion of IFN γ (revised Figure 5D). Following *in vivo* inoculation, basal auto-secretion levels of IFN γ could be lower (due to lower levels of IL-2/IL-15 employed to grow T/NK *in vitro*), thus bringing into account the presence of IFN γ and eventually employed as an auto-feedback loop to keep the high activity of the GCPRE-containing 3-CPRE synthetic promoter in the TME.

It is not clear where ZsGreen comes from in CAR transduced cells for Figure 6- it would be useful if the authors could clarify this. Also degranulation levels are not really convincing. For proof of principle, the authors should at least perform cytotoxicity assays and cytokine releasing assay in order to accurately measure CAR activity and compare this data to a constitutively expressed CAR.

In the original Ms Figure 6D showed the raw data for ZsGreen negative and positive primary T cells. ZsGreen negative primary T cells do not have the construct harboring the CAR

encoded by the synthetic promoter. Thus, CD107a degranulation of ZsGreen-negative T cells (0.97% for unstimulated and 0.99% for stimulated, on JIMT1) represents background levels. To evaluate the effect of the stimulation, we should relate to the ZsGreen positive T cells that are the T cells that do have the 3CPRE promoter-CAR construct. We are now better clarifying it in the text (lines 318-322). Please note that in the revised Ms, we added results with CAR-T cells incubated with additional HER2-positive and negative target cells (HEK293, HeLa, JIMT-1, MCF7, SKOV3, Jurkat as HER2-negative). Revised Fig. 6 show the summary of 3 independent experiments for 6 target cells incubated with non-stimulated and stimulated T cells. Statistical significance of differences in activations indicate that the data is reproducible and significant.

With regard to testing IFN γ secretion by the CAR-T cells, we could not do this assay. Reason: stimulating the T cells harboring 3CPRE promoter with TNF α induce IFN γ secretion (without any target cells) as shown in revised Figure 5D. also, CD107a is considered as a reliable marker for evaluation of cytotoxicity as described here and elsewhere: Aktas, Esin, et al. "Relationship between CD107a expression and cytotoxic activity." Cellular immunology 254.2 (2009): 149-154.).

Authors should further test the activity of the CAR in vivo in order to assess the ultimate functionality of this technology and consolidate the proof of concept. It is again unclear which construct was injected into the mice- did it have the CAR or was it only the RFP670 expressing construct? In any case the expression of RFP670 in Figure 7B is pretty low and it is not possible to determine if this level of expression is functionally significant and will have a biological effect. Thus, the authors should also test CAR T-cell activity in vivo for tumor clearance and show that this activity is tumor specific and not elicited by normal tissues.

The Reviewer is right and we should do all efforts to further expand our studies in this direction. However, we believe that this is out of the scope of this first paper on this technology; before beginning the studies for the revision we pre-discussed with the editors that this task is beyond the scope of the revision and that we cannot fulfill it in 4 months revision time (definitely in the limitations of research work at the universities during the corona era). We definitely plan to do it and we discuss the importance of this stage in the text (lines 431-434).

Did the authors examine other parts of the mouse for activation of CAR? Since the authors present this technology as an approach to overcome off-target toxicity, this should be more thoroughly investigated.

In the original Ms, we compared to inoculation of effector cells into Matrigel for the On-Target-Off tumor cytotoxicity. In the revised Ms, we added new experiments in which we also explored the activation of the 3-CPRE promoter in the effector cells within the blood (revised supplemental Figure 7B and text lines 341-347).

Reviewer #2 (Remarks to the Author):

This manuscript presents a series of vectors in which transgene expression is modulated by exposure to inflammatory cytokines and hypoxia. While conceptually of interest, I have several major reservations about the work.

1. Authors should justify why they selected to make gene expression responsive to IFN γ and TNF α , rather than tumour-associated inhibitory cytokines such as TGF β for example. What levels of these pro-inflammatory cytokines are actually present in typical solid tumors and how does that relate to concentrations of these cytokines used in experiments shown.

It is an important point. Thank you. In the submitted MS, we shortly discuss this issue and cite four references. We now expanded our discussion and added five additional references (numbered 16-20 in the Ms).

Although IFN γ is often represented as antitumor cytokine, IFN γ can efficiently mediate signaling that elicits pro-tumorigenic transformations and promotes tumor progression. Similarly, as a pro-inflammatory cytokine, TNF α is secreted by inflammatory cells, which could be involved in inflammation-associated carcinogenesis. Others and we have previously showed that the presence of IFN γ in the TME could be directly associated with tumor virulence (line 84 ref 16 in the revised Ms). We added 3 additional references (including ours) to better explain our decision to begin our initial studies with IFN γ and TNF α . We also strengthened the discussion for our decision to add hypoxia element, which is a recognized driver in the TME of tumor metastasis (line 84 ref 17-20 in the revised Ms). In accordance, we expanded the text in the discussion (lines 384-388).

2. In contrast to the cytokines chosen, the logic of rendering gene expression inducible in hypoxia is clear, given its prevalence in advanced solid tumors. However, authors should acknowledge others who have attempted something similar previously eg Juillerat et al Sci Rep (2017) 7: 39833.

We thank the reviewer for bringing to our attention this important reference and we are discussing it in the revised Ms (ref 15 and text lines 75-79).

3. Expts in Fig 2 entail the use of a single concentration of cytokine without justification. Dose response and time response curves should be shown in these experiments (e.g. how long to switch on; how long to switch off). Similarly, the performance of the hypoxia-responsive promoter should be evaluated across a spectrum of oxygen tensions and over time.

Figure 4 is now completely revised. In the revised manuscript we show dose response curves of single treatment with IFN γ or with TNF α for the 3-CPRE promoter (the lines of the normoxic; revised Figure 4C and text 260-279). Time response curves for the 3-CPRE promoter are also shown in the new supplemental Figure 5 and in the text 276-279.

4. The number of replicate expts and nature of the error bar should be indicated in the legend to all relevant figures.

In the revised Ms, all legends include no. of replicate experiments and nature of error bars.

5. The performance of the mini-TK promoter alone should be demonstrated throughout the manuscript. This control is also required to assess leakiness of the other promoters when tested in the absence of the corresponding stimulus (e.g. cytokines and/or hypoxia).

Thank you for pointing this. We added this control (only mini-TK with no responsive element) as well as constitutive promoter control in the revised supplemental Figure 2D and in the text lines 223-225 .

6. The experiment shown in Figure 4 seems to be a single example and lacks many controls. Figure 4 is now completely revised. Additional experiments were performed to fully characterize the various combinations of cytokine concentrations. Non-stimulated control (normoxic vs hypoxic) in various combination of cytokine levels, including no cytokine-stimulation at all, is shown in the zero axes-interception point in each of the 12 insets in Figure 4C. IFN γ stimulation alone, over range of cytokines, appears now in the normoxic graph in inset in which TNF α is zero units. Revisions are described in the text lines 265-280.

7. Background production of cytokines such as IFN γ by activated T-cells or NK cells is a clear drawback of the proposed strategy.

Thank you for this important comment. We are definitely exploring now new 3CPRE promoters having promoter elements induced by other cytokines characterizing the tumor microenvironment (e.g TGF β instead of IFN γ). Yet, there is some importance for the IFN γ GCPRE in the context of T and NK cells. The T/NK cells grown *in vitro* auto-saturate the GCPRE element by autocrine secretion of IFN γ (revised Figure 5D). Following *in vivo* inoculation, basal auto-secretion levels of IFN γ could be lower (due to lower levels of IL-2/IL-15 employed to grow T/NK *in vitro*), thus bringing into account the presence of IFN γ and eventually employed as an auto-feedback loop to keep the high activity of the GCPRE-containing 3-CPRE synthetic promoter in the TME.

8. Reproducibility of data in Fig. 6 is unclear.

We previously showed results for JIMT1 and HEK293T target cells. In the revised manuscript we bring results with CAR-T cells incubated with additional HER2-positive and negative target cells (MCF7, SKOV3, HeLa, HEK293, JIMT-1 and Jurkat as HER2-negative). Revised Fig. 6 show the summary of 3 independent experiments for 6 target cells incubated with non-stimulated and stimulated T cells. Statistical significance of differences in activations indicate that the data is reproducible and significant.

9. Therapeutic activity of CAR T-cells is not shown *in vivo*.

The Reviewer is right and we should do all efforts to further expand our studies in this direction. However, we believe that this is out of the scope of this first paper on this technology; before beginning the studies for the revision we pre-discussed with the editors that this task is beyond the scope of the revision and that we cannot fulfill it in 4 months revision time (definitely in the limitations of research work at the universities during the corona era). We definitely plan to do it and we discuss the importance of this stage in the text (lines 431-434).

Minor

1. Describe (tumor?) origin of JIMT1 cells.

Thank you. In the revised Ms we added a reference that fully characterizes the establishment and properties of the cell line (ref. no. 31).

2. Provide nucleic acid sequences of all CARs in supplementary material.

CAR and all CPRE are now listed in supplemental Figure 1.

3. Supplementary Figure 1A does not show expression of IFN γ and TNF α receptors by HEK293T cells.

Thank you noticing this point. Revised supplementary Figure 2A now shows the expression of IFN γ and TNF α receptors by HEK293T cells. In addition, we expanded the Figure to show that these receptors are functional:

(i) MHC-I is induced on HEK293T cells following exposure to IFN γ (supplemental Figure 2B, showing response in titrated cytokine concentrations and in two time points after exposure)

(ii) HEK293T cells transfected with a commercial TNF α reporter responded to TNF α (supplemental Figure 2BC, showing response in titrated cytokine concentrations and in two time points after exposure).

We also describe these results in the text (213-216).

Reviewers' comments:

Reviewer #1 (Remarks to the Author):

Thanks for the revision and responses to all comments. The additional experiments add robustness to the study and, at this point, I have no more additional comment

Reviewer #2 (Remarks to the Author):

This revised manuscript has addressed some of the comments raised but there are a number of significant outstanding issues.

Major

1. In my initial comments, I had stated that "The performance of the mini-TK promoter alone should be demonstrated throughout the manuscript. This control is also required to assess leakiness of the other promoters when tested in the absence of the corresponding stimulus (e.g. cytokines and/or hypoxia)." These data are still not shown other than for mini-TK (without untransduced control) in Fig. S2D. Thus, we don't fully understand how leaky the system is. For example, background reporter activity of the mini-TK promoter and an untransduced T-cell control are not shown in Fig. 5. Similarly in Fig. 6B, we need to see the level of binding of ErbB2-Fc to untransduced T-cells, ZsGreen+ (transduced) T-cells in the uninduced state and ZsGreen+ T-cells following exposure to hypoxia and TNF. Degranulation of MCF7-stimulated T-cells in the uninduced state is higher than baseline (Fig. 6c), which may be suggestive of some leakiness. Analysis of cytokines in supernatant such as IFN γ would help to confirm this point.
2. In regard to possible leakiness, flow cytometric analysis of untransduced NK92 cells should be shown in Fig. 7B. Where does this RFP670 histogram sit in respect of the green matrigel histogram?
3. In my original comments, I had stated that "Background production of cytokines such as IFN γ by activated T-cells or NK cells is a clear drawback of the proposed strategy." Authors now confirm this and should add a comment to indicate this fundamental limitation of the technology.
4. Therapeutic activity of CAR T-cells is not shown in vivo.

Minor

1. Please provide source of NF κ B cherry RFP reporter plasmid
2. Please provide the key to the color scheme in Fig. 4D

Reviewers' comments:

Reviewer #1 (Remarks to the Author):

Thanks for the revision and responses to all comments. The additional experiments add robustness to the study and, at this point, I have no more additional comment

Reviewer #2 (Remarks to the Author):

This revised manuscript has addressed some of the comments raised but there are a number of significant outstanding issues.

Major

1. In my initial comments, I had stated that "The performance of the mini-TK promoter alone should be demonstrated throughout the manuscript. This control is also required to assess leakiness of the other promoters when tested in the absence of the corresponding stimulus (e.g. cytokines and/or hypoxia)." These data are still not shown other than for mini-TK (without untransduced control) in Fig. S2D. Thus, we don't fully understand how leaky the system is. For example, background reporter activity of the mini-TK promoter and an untransduced T-cell control are not shown in Fig. 5. Similarly in Fig. 6B, we need to see the level of binding of ErbB2-Fc to untransduced T-cells, ZsGreen+ (transduced) T-cells in the uninduced state and ZsGreen+ T-cells following exposure to hypoxia and TNF. Degranulation of MCF7-stimulated T-cells in the uninduced state is higher than baseline (Fig. 6c), which may be suggestive of some leakiness. Analysis of cytokines in supernatant such as IFN γ would help to confirm this point.

The question of background expression is indeed an important one. We performed additional experiments thanks to the good advice of reviewer. Figure 2S now include an untransduced control, and another control of mini-TK promoter, and also control constitutive promoter. The data show that background expression by the mini-TK promoter is very low, and it is important to have all controls as reviewer correctly noted. We further revised figures 5-7 to include the un-transduced control (shown as GRAY histograms), as requested.

The comment regarding MCF7 is correct- indeed there is a slightly higher background degranulation with these target cells. The background for each target cell line may differ by multiple factors, including surface activating- or inhibitory ligands and multiple secreted cytokines that may indeed affect the degranulation of our engineered T-cells. In this case, we may hypothesize that the MCF7 might secrete some cytokines that activate our T-cells, and we add specific notion in the text [lines 327-328].

IFN γ secretion by the CAR-T cells was suggested before, but we could not test it separately because the protocol for stimulating the T cells harboring 3CPRE promoter with TNF α already induced IFN γ secretion (without any target cells) as shown in Figure 5D. Importantly, CD107a is a reliable marker for evaluation of cytotoxicity and is a conventional assay (eg:

Aktas, Esin, et al. "Relationship between CD107a expression and cytotoxic activity." Cellular immunology 254.2 (2009): 149-154).

2. In regard to possible leakiness, flow cytometric analysis of un-transduced NK92 cells should be shown in Fig. 7B. Where does this RFP670 histogram sit in respect of the green matrigel histogram?

Thanks for this specific comment - we added the requested FACS histogram to the revised figure 7B (gray histogram). In agreement with our data shown in Figure 5A, the un-transduced cells are having a lower background fluorescence levels, as the reviewer suggested.

3. In my original comments, I had stated that "Background production of cytokines such as IFN γ by activated T-cells or NK cells is a clear drawback of the proposed strategy." Authors now confirm this and should add a comment to indicate this fundamental limitation of the technology.

Thanks again for this constructive specific comment that truly describe a limitation of this system, like any new technology it has advantages and drawbacks. The IFN γ was chosen as a primary signal in this study since it has been reported to have a local impact on cytotoxic cells in the TME that may be further increased by autocrine secretion, possibly making an interesting positive feed-forward loop. As requested by the reviewer, we added a specific comment in the discussion to help the readers [lines 434-437].

4. Therapeutic activity of CAR T-cells is not shown in vivo.

This is true, and we had already discussed this point in the previous correspondence. Truly, activity of CAR T-cells in vivo is interesting and opens many options for multiple future studies. We wish to publish our novel system and provide for many groups the opportunities to further exploit it, and develop multiple in vivo applications.

Minor

1. Please provide source of NFkB cherry RFP reporter plasmid

Thanks for this correction. We added the plasmid source in the method chapter [lines 117-118].

2. Please provide the key to the color scheme in Fig. 4D

Thanks for this correction. The color key is added as requested in Fig. 4D right below the heatmap.

REVIEWERS' COMMENTS:

Reviewer #2 (Remarks to the Author):

Thank you for your further rebuttal. I am satisfied that the manuscript is now suitable for publication.